# The Effect of the Yingyangbao Complementary Food Supplement on the Nutritional Status of Infants and Children: A Systematic Review and Meta-Analysis

**DOI:** 10.3390/nu11102404

**Published:** 2019-10-08

**Authors:** Zhihui Li, Xinyi Li, Christopher R. Sudfeld, Yuning Liu, Kun Tang, Yangmu Huang, Wafaie Fawzi

**Affiliations:** 1Department of Global Health and Population, Harvard T.H. Chan School of Public Health, Boston, MA 02115, USA; zhl922@mail.harvard.edu (Z.L.); crsudfeld@gmail.com (C.R.S.); lynpkuhsc@126.com (Y.L.); mina@hsph.harvard.edu (W.F.); 2Department of Epidemiology, Harvard T.H. Chan School of Public Health, Boston, MA 02115, USA; 3Department of Nutrition, Harvard T.H. Chan School of Public Health, Boston, MA 02115, USA; 4Research Center for Public Health, School of Medicine, Tsinghua University, Beijing 100084, China; tangk@mail.tsinghua.edu.cn; 5Department of Global Health, School of Public Health, Peking University, Beijing 100191, China

**Keywords:** Yingyangbao, food supplement, infant and children, hematological status, anthropometric status

## Abstract

Yingyangbao (YYB) is a nutrient-dense complementary food supplement for infants and young children in China. There has been considerable interest and research on the potential effects of YYB on hematological and anthropometric outcomes in China, but limited effort has been made to consolidate and synthesize the evidence to inform the research and policy agendas. Eight English databases and three Chinese databases were searched from January 2001 to June 2019 to identify YYB intervention studies. A total of 32 quasi-experimental, post-only, concurrent-control studies or pre-post studies were identified, and 26 were included in the meta-analyses. A pooled analysis of post-only studies with concurrent-control determined that YYB was associated with an increase of 4.43 g/L (95% confidence interval (CI) 1.55, 7.30) hemoglobin concentration, 2.46 cm (CI 0.96, 3.97) in height, and 0.79 kg (CI 0.25, 1.32) weight in infants and young children. YYB was also associated with reductions in the prevalence of anemia (risk ratio (RR) = 0.55; 95% CI: 0.45, 0.67), stunting (RR = 0.60; 95% CI: 0.44, 0.81), and underweight (RR = 0.51; 95% CI: 0.39, 0.65). Overall, YYB was found to be associated with improved hematological and anthropometric indicators among infants and young children in China; however, randomized trials are needed to causally assess the efficacy of YYB due to the inherent risk of bias in existing quasi-experimental studies; rigorous implementation and cost-effectiveness evaluations are also needed.

## 1. Introduction

China is home to 83 million children under 5 years of age, accounting for 13% of the global population in that age category in 2017 [1,2]. Despite rapid economic growth, China still has large number of undernourished children. In 2013, around seven-million Chinese children were stunted and two-million were underweight [3,4]. Anemia also remains a severe problem in China, particularly in rural areas: 28% of rural children between 6 and 12 months old and 21% between 13 and 24 months are estimated to be anemic [5].

To improve child nutritional status in rural China, Yingyangbao (YYB) was developed as a nutrient-dense complementary food supplement for infants and young children 6–36 months old. The base of YYB is full fat soybean powder with multiple additional micronutrients, including calcium, iron, zinc, vitamin B1, vitamin B2, vitamin B12, vitamin A, and vitamin D. The composition of YYB varies slightly by the manufacturing company [6,7,8,9,10,11], and in some formulations, folic acid, and omega-3 or omega-6 fatty acids are added. YYB also provides calories (usually around 50 kcal per sachet) and protein (3 g per sachet) [6,7,8]. 

The first YYB study was launched in 2001 in five poor counties of Gansu Province, China, among children 4–12 months old [8]^8^. After the pilot, the provision of YYB has gradually expanded to other poor rural areas of China and regions affected by natural disasters. Early YYB programs provided YYB to children from 4 months old due to national-exclusive feeding guidelines that were later changed to 6 months in 2007 [12]. Consequently, YYB programs implemented post 2007 were primarily among children 6–36 months old. In 2011, the Chinese government invested ¥100 million (USD 16 million) in the YYB program to cover 300,000 children in 100 counties across 10 provinces [13]. This investment was expanded to ¥500 million (USD 81 million) for 1.4 million children across 21 provinces. In 2017, the Chinese government issued a national nutrition strategy for 2017–2030 that prioritized the nutritional status of children in their first 1000 days, and YYB was included as a key intervention for rural and poor populations [14]. In 2019, the Chinese government launched the “Upgrade of YYB Plan” which will provide YYB to infants and children in 823 poor counties [15]. 

Multiple quasi-experimental research studies have estimated the impact of the YYB program on anthropometric indicators and hematological parameters [6,7,8,9,16,17]; however, the results have been heterogeneous. For example, Wang J et al. (2017) found that YYB significantly reduced stunting prevalence; however, another study by Zhang Y et al. (2016) reported no significant effect [6,7]. Two prior meta-analyses have been conducted to estimate the effects of YYB program [9,17]; however, they did not include several recently published studies and incorrectly included a single study multiple times in pooled analyses [9,17,18]. In addition, the prior reviews did not consider differences in study design or selection of control participants, and they did not assess the full range of anthropometric and hematologic indicators. Therefore, this systematic review was conducted to address these limitations. The authors also summarize the state of the evidence on YYB, and propose the next steps for advancing research and policy.

## 2. Materials and Methods

This systematic review of YYB was performed according to the Preferred Reporting Items for Systematic Reviews and Meta-analysis (PRISMA) guidelines [19]. This article sought to examine the effectiveness of providing YYB to infants and children on all health and nutrition outcomes assessed in published studies, including: (1) anthropometric outcomes including height (cm), weight (kg), height-for-age z score (HAZ), weight-for-age Z score (WAZ), weight-for-height Z score (WHZ), stunting prevalence (defined as HAZ ≤ −2 standard deviations [SD]), underweight prevalence (defined as WAZ ≤ −2 SDs), and wasting prevalence (defined as WHZ ≤ −2 SDs); (2) hematological parameters, including hemoglobin (Hb) concentration (g/L) and anemia prevalence (defined as Hb concentration <110 g/L); (3) child developmental outcomes including intelligence quotient and developmental quotient; and (4) disease prevalence, including the prevalence of diarrhea and the prevalence of respiratory infections in prior two-weeks.

### 2.1. Study Selection

Studies were eligible for inclusion if they: (1) included YYB as an intervention, (2) compared the effect of YYB between intervention group and a control group (quasi-experimental study), or (3) assessed the prevalence of health and nutrition outcomes between before and after YYB implementation (pre-post study). Studies were excluded if they were not academic studies, implemented a non-YYB complementary feeding intervention (like a soybean powder based, or micronutrient supplementation without energy), did not target children under 5 years old, or did not contain health or nutrition outcomes. The YYB program has not been implemented outside China, and therefore, this review only included studies published after 2001, which is when the first YYB study was carried out in China. 

### 2.2. Data Sources and Search Strategy

A total of 11 databases were searched, including eight English databases covering nutrition, social sciences, and rural development: Cochrane Library, Medline, Econlit, PubMed, Embase, Web of Science, CAB Abstracts (EBSCO, Ipswich, MA, USA), CINAHL, and three Chinese databases including Wanfang data, China National Knowledge Infrastructure, and VIP—over the period from January 2001 to June 2019. Separate search strategies consisting of a combination of free text words (tw), words in titles/abstracts (tiab) and medical subject headings (mesh) for interventions, participants, and study designs were developed, and then combined by using “AND.” The following search strategy was run across in PubMed and tailored to each database when necessary: (“nutritional package” (tiab) or “micronutrient package” (tiab) or “micronutrient supplementation” (tiab) or “multi-micronutrient supplementation” (tiab) or “multi-nutrient powder” (tiab) or “complementary supplementation” (tiab) or “micronutrient supplement” (tiab) or “multi-micronutrient supplement” (tiab) or “complementary supplement” (tiab) or “Yingyangbao” (tiab) or “YYB” (tiab) or “Ying Yang Bao” (tiab) or “nutritional intervention” (tiab) or “Nutrition Policy” (Mesh) or “Dietary Supplements” (Mesh)) and (“China” (tiab)) and (“infant” (tiab) or “infancy” (tiab) or “child” (tiab) or “children” (tiab)) and 1 January 2001–1 June 2019. No language restriction was placed. The reference lists of included studies, and previous systematic reviews were also searched to identify additional eligible studies.

### 2.3. Data Extraction and Assessment of Methodological Quality

Two reviewers (X.L. and Y.L.) independently extracted data using a pilot tested data extraction form. Information from each study on study design, setting, participants, intervention, control, and outcomes were collected. Intervention details included the composition of YYB, the frequency of taking YYB (per day, or per 10 days), and intervention duration in months. No individually or cluster-randomized controlled trials were identified but multiple quasi-experimental studies were included. Two types of quasi-experimental study designs were distinguished: those that had a non-randomized control group (post-only with concurrent-control study) and those that compared outcomes before and after the YYB intervention in the same communities. Mean and standard deviation values were collected for continuous outcomes, as were the number of cases and non-cases in each arm for binary outcomes. 

The methodological index for non-randomized studies (MINORS) [20] was used to assess the quality of post-only studies with concurrent-control and pre-post studies. The quality of post-only studies with concurrent-control was assessed with 12 criteria: (1) the stated aim of the study, (2) consecutive inclusion of participants, (3) prospective collection of data, (4) appropriate endpoints, (5) unbiased evaluation of endpoints, (6) appropriate follow-up period, (7) loss to follow-up ≤5%, (8) appropriate control group, (9) contemporary groups, (10) baseline equivalence of groups, (11) prospective calculation of sample size, and (12) statistical analyses adapted to the study design. The first seven criteria were considered when assessing the quality of pre-post studies, with each criterion assigned a score of 0 to 7. The potential range of total score was 0 to 84 for studies with concurrent-control, and 0 to 49 for pre-post studies. Articles were categorized into high, moderate, and low quality based on total score tertiles. Concurrent control studies with scores higher than 53 were defined as high quality; those with a score of 50–53 as moderate-quality; and those with scores of lower than 50 as low-quality. As for pre-post studies, studies with scores higher than 35, in the range of 31–35, and less than 31, were categorized as high, moderate, and low quality, respectively. Quality scores are presented in Appendix A.

### 2.4. Statistical Analyses

All analyses were stratified by study design due to differences in risk of bias. For post-only with concurrent-control YYB studies, the comparison of outcomes was between intervention and control groups at the end of the intervention period, since pre-intervention assessments or difference-in-difference estimates were not provided in most studies. For pre-post studies, outcomes were compared before and after the YYB intervention. Estimates were pooled separately for concurrent control studies and pre-post studies for outcomes where three or more studies provided estimates. Between-study heterogeneity was assessed by I^2^ statistics. Random effects meta-analysis was used if substantial heterogeneity was present, indicated by *p* < 0.1 in χ^2^ test and an I^2^ > 50%, otherwise the fixed effects were applied [21]. Pooled mean differences (MD) were reported for continuous outcomes and risk ratios (RR), and risk differences (RD) for binary outcomes and their corresponding 95% confidence intervals (CIs). 

Two pre-specified subgroup analyses were conducted to explore sources of heterogeneity: (1) control type (active control; i.e., providing rice flour with the same energy as YYB; or a blank control where no intervention was given) and (2) intervention duration (≤12 months or >12 months). 

To detect the robustness of the results, sensitivity analyses were also conducted by excluding studies with low quality. When an adequate number of studies (number of studies ≥ 10) were available [21], the potential for publication bias was assessed by the visual inspection of funnel plots for asymmetry and through Egger’s linear regression tests. Stata (version 14.0, STATA, College Station, TX, USA) and Cochrane Review Manager (RevMan) (version 5.3 Cochrane Community, London, UK) were used for all analyses. 

## 3. Results

### 3.1. Literature Search and Selected Studies

A total of 3092 reports were identified in our systematic search of the literature (Figure 1). Seventy-seven of them were identified as potentially relevant reports for review after screening the titles and abstracts. A full text review of these papers yielded 32 studies for inclusion in the systematic review, including 18 post-only studies with concurrent-control and 14 pre-post studies. Only outcomes reported in three or more studies were examined in meta-analysis, and as a result, 26 studies were in the meta-analyses (13 post-only studies with concurrent-control and 13 pre-post studies).

Table 1 and Table 2 present the characteristics of included post-only studies with concurrent-control and pre-post studies, respectively. Among the 18 concurrent control studies, 17 provided YYB on a daily basis (one sachet per day) and one study provided one sachet of YYB every 10 days; 11 studies had a blank control and seven included isocaloric rice flour as the active control. All the 14 pre-post studies provided one sachet of YYB per day. The composition of YYB varied slightly across studies, with three major types (referred to as YYB-A, YYB-B, and YYB-C). Table 1 and Table 2, and Appendix A, present composition details for the YYB supplement in each study. A majority of the included studies were conducted among children aged 6–24 months old.

### 3.2. Continuous Hematologic and Anthropometric Outcomes

The results of meta-analyses on Hb concentration, weight, and height are summarized in Figure 2 and Table 3. A pooled analysis of seven post-only studies with concurrent-control on Hb concentration indicated that YYB increased Hb concentrations by 4.43 g/L (95% CI: 1.55, 7.30; *p* = 0.003) compared to control; however, it is important to note there was large heterogeneity in effect sizes across studies (I^2^ = 96%, *p* < 0.001). Consistently, the pooled results of seven pre-post studies indicated a 6.58 g/L (95% CI: 2.71, 10.45; *p* < 0.001) increase in Hb concentration after the YYB intervention (Table 3 and Appendix A). 

The effect of YYB on children’s height and weight was assessed in six post-only studies with concurrent-control. Children’s height was 2.46 cm greater (95% CI: 0.96, 3.97; *p* = 0.001) and weight was on average 0.79 kg heavier (95% CI: 0.25, 1.32; *p* = 0.004) in the YYB groups compared to control groups. Heterogeneity was also high in height and weight meta-analyses (I^2^ = 94%, *p* < 0.002, and I^2^ = 97%, *p* < 0.001, respectively). The pre-post studies showed comparable results; children after the YYB intervention were on average 2.46 cm (95% CI: 1.35, 3.58; *p* < 0.001) taller and 0.72 kg (95% CI: 0.33, 1.10; *p* < 0.001) heavier than before intervention (Table 3, and Appendix A).

Table 3 and Appendix A present the effect of YYB on HAZ, WHZ, and WAZ. Only three post-only studies with concurrent-control reported effect estimates on HAZ, and in pooled analyses there was no effect of YYB on continuous HAZ (mean difference: −0.03; 95% CI: −0.68, 0.62; *p* = 0.94). Yet, the pooled result from five pre-post studies indicated that HAZ significantly increased by a large 0.24 SD (95% CI: 0.10, 0.39; *p* < 0.001) after YYB intervention with moderately high heterogeneity (I^2^ = 70%, *p* = 0.01). Results from three concurrent control studies showed that children in the YYB group had 0.27 of a SD (95% CI: 0.09, 0.45; *p* = 0.003) higher WHZ than children in the control group. There was a similar point estimate of 0.28 SD in pre-post studies, but the results were not statistically significant. In terms of WAZ, there was no effect of YYB in the three concurrent control studies or the five pre-post studies (Table 3 and Appendix A).

### 3.3. Anemia, Stunting, Underweight, and Wasting

Figure 3 and Table 4 present the overall effect of YYB program on anemia prevalence. Eleven post-only studies with concurrent-control were included in the pooled analyses, and a 45% reduction in the risk of anemia (RR = 0.55; 95% CI: 0.45, 0.67; *p* < 0.001) was observed with high heterogeneity in effect estimates (I^2^ = 84%, *p* < 0.001). Pooled analyses of 13 pre-post studies similarly showed that anemia prevalence was reduced by 42% after the YYB intervention (RR = 0.58; 95% CI: 0.50, 0.68; *p* < 0.001) (Appendix A).

A pooled analyses of seven post-only studies with concurrent-control found that children’s risk of stunting in the YYB groups was reduced by 40% (RR = 0.60; 95% CI: 0.44, 0.81; *p* < 0.001) compared to the control groups; there was high heterogeneity in this analysis (I^2^ = 71%, *p* = 0.002). Consistently, pooled analysis of 10 pre-post studies found that YYB was associated with a 25% (RR = 0.75; 95% CI: 0.60, 0.95; *p* = 0.02) lower risk of being stunted after the intervention (Table 4 and Appendix A). Pooled analyses showed that YYB was associated with lower risk of underweight (Table 4 and Appendix A). In terms of wasting, YYB was associated with 52% (RR = 0.48; 95% CI: 0.32, 0.70; *p* < 0.001) lower risk of being wasting in the five post-only with concurrent-controll studies, yet the seven pre-post studies did not yield any significant effect (Table 4 and Appendix A).

The summary risk differences and 95% CIs for categorical outcomes were also calculated (Appendix A). Overall, the findings for anemia, stunting, underweight, and wasting were consistent with the relative risk analyses. 

### 3.4. Subgroup and Sensitivity Analysis

Subgroup analyses were conducted among post-only with concurrent studies by the type of control (active or blank). Pooled analyses of blank control studies determined that YYB was associated with a 5.54 g/L (95% CI: 1.46, 9.61; *p* = 0.001) increase in Hb concentration (I^2^ = 0%). The subgroup of active control studies also found YYB was associated with higher Hb concentration, but the effect size was 1.80 g/L (95% CI: 0.77, 2.83; *p* = 0.008) (I^2^ = 97%). The increase of Hb concentration in blank control group studies was significantly greater than that found in active control group studies (*p* < 0.001).

In terms of anemia prevalence, nine studies were with blank controls and two studies with active controls. Pooled analyses showed that risk of anemia was significantly lower in both blank control and active control studies (blank control: RR = 0.64; 95% CI: 0.48, 0.87; *p* = 0.004; I^2^ = 87%; active control: RR = 0.52; 95% CI: 0.41, 0.66; *p* < 0.001; I^2^ = 0%), and the risk reduction was not significantly different between two control types (*p* = 0.93). 

No evidence of modification by intervention duration (≤12 months or >12 months) on YYB effect was observed (Appendix A). In addition, results remained consistent when excluding five concurrent-controlled and four pre-post studies that were determined to be of low quality (Appendix A). 

Funnel plots (Appendix A) and Egger’s tests for analyses of anemia prevalence did not suggest the presence of publication bias for concurrent control studies (*p* = 0.493 for Egger’s test) or pre-post studies (*p* = 0.388).

## 4. Discussion

In this study, YYB complementary food supplement was found to be associated with significant increases in children’s Hb concentration, height, and weight, and significant reductions in children’s risk of anemia and stunting, in quasi-experimental studies conducted in China. Subgroup analyses by control group and with sensitivity analyses limited to high-quality studies were consistent with the main results. 

Consistent with other studies of multi-micronutrient supplementation (e.g., Sprinkles, Foodlet, and lipid-based nutrient supplements (LNS)) [9,50,51,52,53,54,55], YYB resulted in a significant improvement in children’s hematological outcomes in the current study. As a previous study suggested [56]^56^, the iron composition of the supplement might be the major contributor to reducing anemia’s prevalence and increasing Hb concentration. Despite the importance of iron supplementation, the unpleasant taste of iron has been widely discussed. A study on the YYB program showed that 45.3% of the children did not like the taste of YYB, which may have affected adherence to YYB [6]. Sprinkles have reduced the metallic taste by encapsulating iron within a thin lipid layer to prevent the iron from interacting with food [53]. As a result, the improved taste may improve adherence to and the impact of YYB. 

Previous studies on protein-free multi-micronutrient supplements, such as Sprinkles, Foodlets, and drops, found small or no effects on children’s weight and risk of underweight [52,57,58]. However, YYB appeared to have large effects on weight-related indicators; post-only studies with concurrent-control found a 0.8 kg increase in weight, a 0.27 SD increase in WAZ, and a 49% decrease in the risk of underweight prevalence. Nevertheless, our finding is generally consistent with studies on lipid-based nutrient supplements (LNS), which additionally provide calories, proteins, and fatty acids. LNS refers to a range of products wherein the majority of energy is provided by lipids, including products like ready-to-use therapeutic foods (RUTF) and small-quantity LNSs that provide <110 kcal/day [55]. Yet the effect size of YYB on weight-based outcomes in our meta-analysis appear to be much larger than LNS. A recent meta-analysis of RCTs found LNS reduced the risk of underweight by 15%, compared to a more than a 40% reduction found in our YYB meta-analysis of quasi-experimental studies [55]. Similarly, this study found YYB to have a larger effect on stunting reduction than LNS (20% compared to 7%) [55]. The difference in the effects of YYB and LNS is notable, particularly considering LNS provides greater calories (usually about 100–1000 kcal per day or 50%–100% of daily energy requirements)) compared to YYB (~50 kcal per day, 5% to 25% of daily energy requirements) [55,59,60]. The larger effect of YYB over LNS might be attributable to the differences in the study designs and corresponding risk of bias and the study populations. LNS trials were primarily for children with moderate to acute malnutrition [61], while YYB studies were largely among all children in a community. Therefore, YYB may not only improve growth for undernourished children, but also prevent child malnutrition in the general population, which could have contributed to the larger effect at population-level. 

There are also some concerns with regards to use of LNS. Large quantity LNS has raised substantial concern about the adverse effects on excessive weight gain that may affect later-life, non-communicable disease risk in the future, particularly when the product was used for prevention purposes [62]. Small-quantity LNS provides a lower calorie quantity (∼110  kcal per day) and the evidence based on body composition and long term outcomes is evolving [60]. In addition, sugar is added to LNS primarily to increase palatability and there has been some concern in regard to fostering infant preferences for sweet foods over breastmilk [60]. An additional consideration is that the cost per serving of YYB is believed to be lower than small-quantities of LNS [63]. 

In addition to the effects on hematologic and nutritional outcomes, several papers have examined the association of YYB with additional child health and development outcomes. Two studies that examined developmental outcomes found that YYB program significantly improved intelligence quotient and gross motor development [23,27]. YYB contains macronutrients and micronutrients (e.g., protein, iron, zinc, etc.) that may be critical for brain development [64,65]. An additional study investigated the effect of YYB on childhood morbidities and found that the prevalence of respiratory infection and diarrhea was reduced after a 12-month YYB intervention [24]^24^.

This review has several limitations: First, both concurrent control and pre-post quasi-experimental studies are observational in nature and are at risk of confounding and other biases. As a result, randomized trials are needed to determine causal effects. Second, the small number of studies for several indicators, reduces the power of analysis to examine the effect on anthropometric Z-scores, and limits our ability to do meta-analyses on developmental outcomes or disease prevalences. Third, secular improvement in health and nutrition is of particular risk in pre-post studies. However, the estimates of effects in pre-post studies were generally in line with those from post-only studies with concurrent-control. Fourth, significant heterogeneity in estimated effects was found for most hematologic and anthropometric outcomes, which suggests that the magnitude of the effect of YYB may differ by setting. 

Our findings suggest that YYB may improve hematological outcomes and growth status during infancy and early childhood. YYB is gaining attention both within China and from the global nutrition community [66], particularly, since complementary feeding interventions are included in the WHO Global Nutrition Targets 2025 that call for all countries to reduce the number of children under five who are stunted by 40% from 2015 levels [67]. Despite evidence indicating the potential benefits of YYB, there is clearly a need for more robust evidence, including randomized control trials and rigorously implemented evaluations. In addition, exploring the use of YYB outside of China, including assessments of acceptability, the composition of the product, and impacts, are warranted. In settings in China where YYB is already deployed, future research should be conducted to identify gaps in program implementation, and explore strategies to deliver the YYB with optimal effective coverage. 

## Figures and Tables

**Figure 1 nutrients-11-02404-f001:**
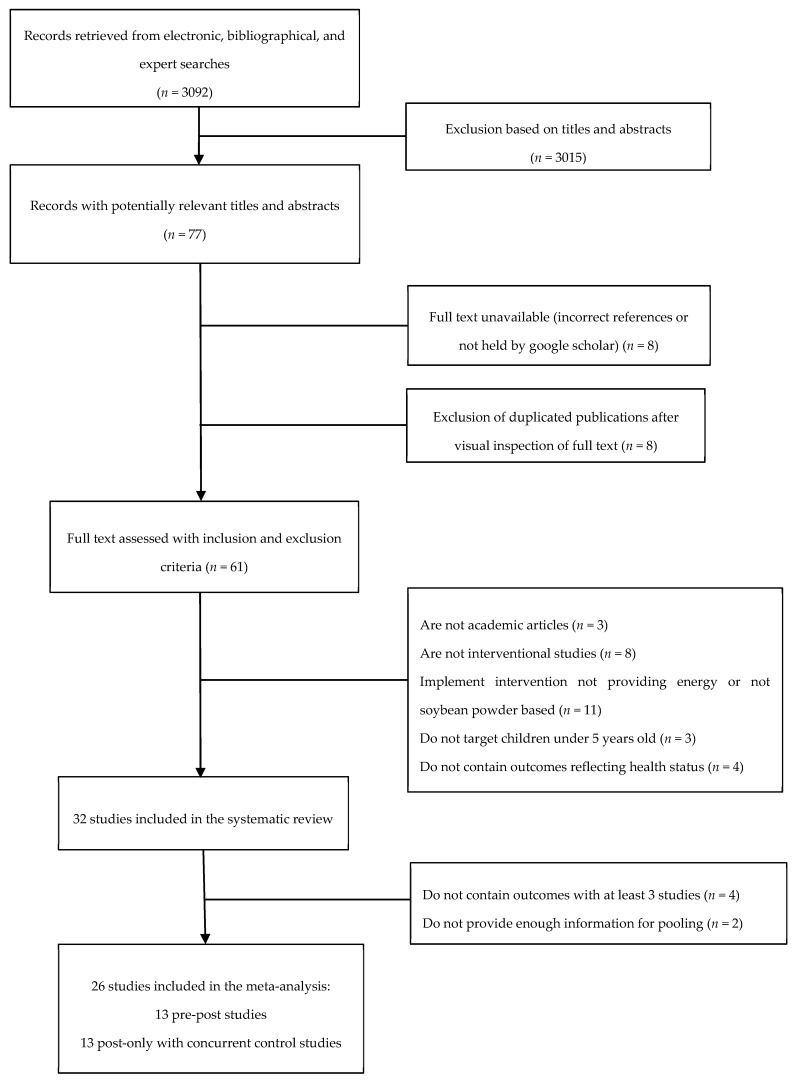
Flow diagram of search and selection process.

**Figure 2 nutrients-11-02404-f002:**
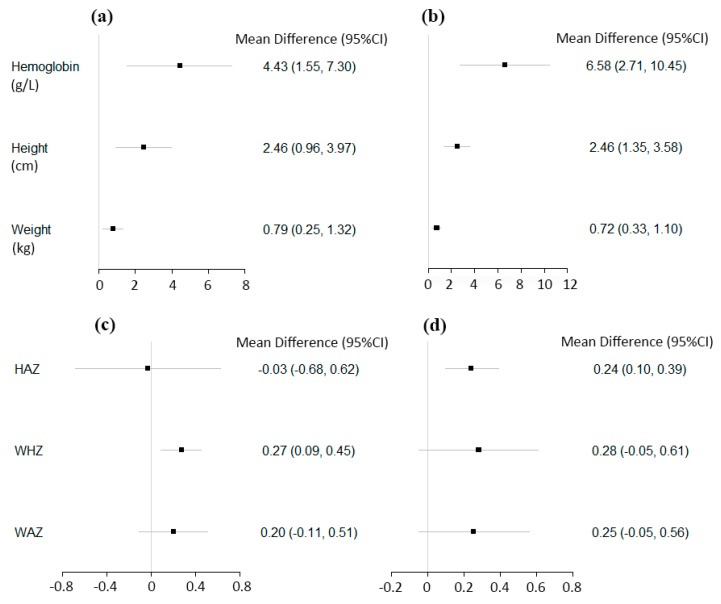
The effect of Yingyangbao (YYB) on continuous hematologic and anthropometric outcomes among post-only studies with concurrent-control (**a**,**c**), and pre-post studies (**b**,**d**) in China.

**Figure 3 nutrients-11-02404-f003:**
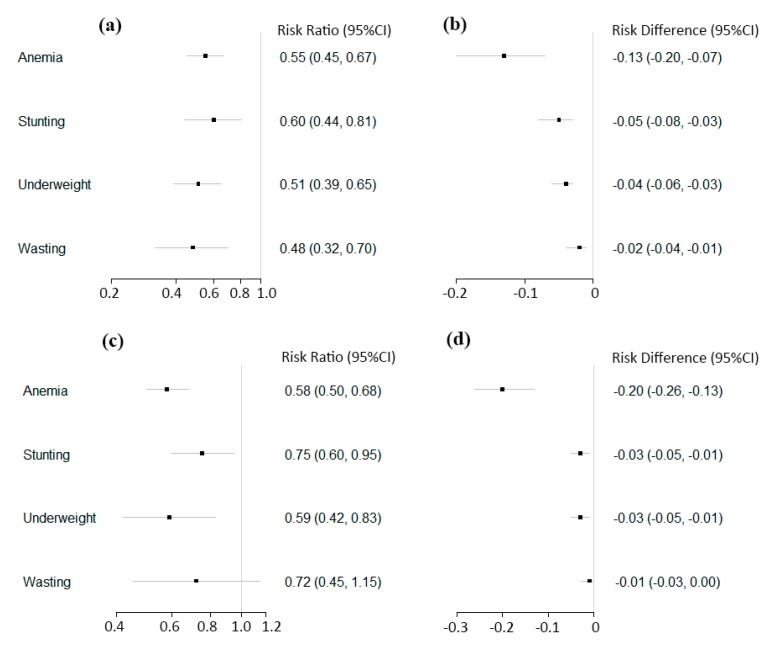
The effect of YYB on the prevalence of anemia and child anthropometric outcomes in China among post-only studies with concurrent-control using risk ratios (**a**) and risk differences (**b**), and among pre-post studies, using risk ratios (**c**) and risk differences (**d**).

**Table 1 nutrients-11-02404-t001:** Characteristics of post-only studies with concurrent-control.

First Author, Year	Target Age (Months)	Intervention Group ^1^	Control Group	Intervention Duration (Months)	Age When Outcomes Measured (Months)	Outcomes Measured ^2^
Shuhua Ni, 1995 [22]	5–13	2 counties; *n* = 76; YYB-C	2 counties; *n* = 77; Blank control	8 months	13	Height, weight, Kaup index, Vervaeck index, urine hydroxyproline index
Yuying Wang, 2004 [10]	4–12	5 counties; *n* = 670; YYB-B	5 counties; *n* = 307; Energy matched rice flour	12 months	16–24	Hb concentration and prevalence of anemia
Yuying Wang, 2006 [23]	4–12	5 counties; *n* = 232; YYB-B	5 counties; *n* = 116; Energy matched rice flour	Until 24 months of age for each child	24	Developmental quotient
Dongmei Yu, 2007 [24]	4–12	5 counties; *n* = 978; YYB-B	5 counties; *n* = 500; Energy matched rice flour	12 months	16–24	Prevalence of diarrhea and respiratory infection in past two weeks
Yuying Wang, 2007 [25]	4–12	5 counties; *n* = 978; YYB-B	5 counties; *n* = 500; Energy matched rice flour	Until 24 months of age	24	WAZ, HAZ
Yuying Wang, 2009 [26]	6–12	5 counties; *n* = 978; YYB-B	5 counties; *n* = 500; Energy matched rice flour	12 months	18–24	Hb concentration and prevalence of anemia
Chunming Chen, 2010 [27]	4–12	5 counties; *n* = 232; YYB-B	5 counties; *n* = 116; Energy matched rice flour	Until 24 months of age	24	Development quotient, intelligent quotient
Zhifeng Fang, 2010 [28]	6–24	3 counties; *n* = 146; YYB-A	3 counties; *n* = 107; Blank control	6 months	12–30	Prevalence of stunting, underweight, wasting and anemia
China Development and Research Foundation, 2011 [29]	6–24	6 counties; *n* = 1034; YYB-A	6 counties; *n* = 449; Blank control	24 months	24–36	Prevalence of underweight, stunting and wasting, Hb concentration and anemia prevalence
Wenli Zhao, 2012 [30]	0–60	2 counties; *n* = 676; YYB-A	2 counties; *n* = 536; Blank control	12 months	12–72	Height, weight, prevalence of stunting, underweight and wasting, Hb concentration and prevalence of anemia
Songli Fan, 2013 [31]	6–24	27 counties; *n* = 113; YYB-A	27 counties; *n* = 328; Blank control	3 months	9–27	Hb concentration and prevalence of anemia
Wenhao Li, 2013 [32]	4–30	3 counties; *n* = 146; YYB-A	3 counties; *n* = 146; Blank control	3 months	7–33	Height, weight, anemia prevalence, and Hb concentration
Lingyun Ren, 2013 [33]	6–11	One county; *n* = 76; YYB-A	One county; *n* = 78; Blank control	Until 24 months of age for each child	24	Height, weight, WAZ, HAZ, prevalence of stunting, underweight and wasting, Hb concentration and prevalence of anemia
Shangming Li, 2014 [34]	0–36	One county; *n* = 387; YYB-A	One county; *n* = 240; Blank control	Until 36 months of age for each child	36	Prevalence of stunting, underweight and anemia
Qin Hu, 2016 [35]	6–36	One county; *n* = 589; YYB-A	One county; *n* = 300; Blank control	6 months	12–42	Height, weight, head circumference, Hb concentration and prevalence of anemia and rickets
Xiaoting Ding, 2016 [36]	6–18	3 counties; *n* = 483; YYB-A	3 counties; *n* = 248; Blank control	6 months	12–24	Height, weight, WAZ, HAZ, WHZ, Hb concentration, prevalence of underweight, wasting, stunting and anemia
Yanfeng Zhang, 2016 [7]	6–23	One county; *n* = 2186; YYB-A	One county; *n* = 760; Blank control	Until 24 months of age for each child	6–23	Prevalence of stunting and anemia
Shuai Li, 2017 [37]	6–24	One county; *n* = 450; YYB-A	One county; *n* = 450; Blank control	12 months	18–36 months	Height, weight, WAZ, HAZ, WHZ, Hb concentration, prevalence of underweight, wasting, stunting and anemia

^1^ Composition of YYB-A, B, C presented in Appendix A. ^2^ Abbreviations: YYB, Yingyangbao; T, intervention group; C, control group; WAZ, weight-for-age z score; HAZ, height-for-age z score; WHZ, weight-for-height z score; Hb, hemoglobin.

**Table 2 nutrients-11-02404-t002:** Characteristics of pre-post studies.

First Author, Year	Target Age (Months)	Intervention ^1^	Intervention Duration	Age When Outcomes Measured (Months)	Outcomes Measured ^2^
Hong Shen, 2011 [38]	0–36	One county; Pretest: *n* = 143 Posttest: *n* = 148; YYB-A	8 months	8–44	Prevalence of anemia
Jing Sun, 2011 [39]	6–24	2 counties; Pretest: *n* = 226 Posttest: *n* = 221; YYB-A	20–24 months of age for each child	6–24	Prevalence of anemia
Lijuan Wang, 2011 [40]	6–29	One county; Pretest: *n* = 257 Posttest: *n* = 253; YYB-A	15 months	6–29	Height, weight, WAZ, HAZ, prevalence of stunting and underweight, Hb concentration and prevalence of anemia
Caixia Dong, 2012 [41]	6–18	One county; Pretest: *n* = 314 Posttest: *n* = 242; YYB-A	18 months until 24 months of age for each child	6–24	WAZ, HAZ, WHZ, prevalence of underweight, stunting and wasting, Hb concentration and anemia prevalence
Linjiang Wang, 2012 [42]	6–24	One county; Pretest: *n* = 327 Posttest: *n* = 300; YYB-A	18 months until 24 months of age for each child	6–24	WAZ, HAZ, Hb concentration
Lixiang Li, 2012 [43]	6–24	One county; Pretest: *n* = 327 Posttest: *n* = 307; YYB-A	6 months	6–24	Height, weight, WAZ, HAZ, prevalence of stunting, underweight and wasting, Hb concentration and prevalence of anemia
Zengkang Xu, 2012 [44]	6–24	One county; Pretest: *n* = 327 Posttest: *n*-300; YYB-A	18 months	6–24	Heigh, weight, WAZ, HAZ, WHZ, Hb concentration, and prevalence of underweight, wasting, stunting and anemia
Zuyang Liu, 2013 [45]	6–24	5 counties; Pretest: *n* = 659 Posttest: *n* = 506; YYB-A	18 months until 24 months of age	6–24	Prevalence of stunting, underweight, wasting and anemia
Jianhong Qin, 2014 [46]	6–24	One county; Pretest: *n* = 159 Posttest: *n* = 206; YYB-A	12 months until 24 months of age for each child	6–24	Prevalence of underweight and stunting, Hb concentration and anemia prevalence, prevalence of diarrhea and respiratory infection
Junsheng Huo, 2015 [47]^47^	6–23	8 counties; Pretest: *n* = 1290 Posttest: *n* = 1040; YYB-A	18 months until 24 months of age for each child	6–23	Hb concentration and prevalence of anemia
Qiannan Zhang, 2015 [9]^9^	6–23	2 counties; Pretest: *n* = 596 Posttest: *n* = 589; YYB-A	12 months until 24 months of age for each child	6–23	Height, weight, WAZ, HAZ, prevalence of stunting, underweight and wasting, Hb concentration and prevalence of anemia
Qiujing Jiang, 2016 [48]	6–24	3 counties; Pretest: *n* = 596 Posttest: *n* = 589; YYB-A	12 months until 24 months of age for each child	6–24	Height, weight, prevalence of stunting and underweight, Hb concentration and prevalence of anemia
Jie Wang, 2017 [6]	6–23	3 counties; Pretest: *n* = 823 Posttest: *n* = 693; YYB-A	18 months until 24 months of age for each child	6–23	Prevalence of underweight, stunting, wasting and anemia, micronutrient status
Ping Wu, 2017 [49]	6–24	One county; Pretest: *n* = 156 Posttest: *n* = 156; YYB-A	24 months until 24 months of age	6–24	Prevalence of stunting, underweight, wasting and anemia

^1^ Composition of YYB presented in Appendix A. ^2^ Abbreviations: YYB, Yingyangbao; T, intervention group; C, control group; WAZ, weight-for-age z score; HAZ, height-for-age z score; WHZ, weight-for-height z score; Hb, hemoglobin.

**Table 3 nutrients-11-02404-t003:** The effect of YYB on children’s hemoglobin concentration and continuous, anthropometric outcomes in China.

**Post-Only Studies with Concurrent-Control**	**Number of Studies**	**Total Intervention Participants**	**Total Control Participants**	**Mean Difference (95% CI)**	***p*-Value for Summary Effects**	**I^2^ for Heterogeneity (%)**
Hemoglobin concentration (g/L)	7	2810	1714	4.43 (1.55, 7.30)	0.003	96
Height (cm)	6	2061	1689	2.46 (0.96, 3.97)	0.001	94
Weight (kg)	6	2061	1689	0.79 (0.25, 1.32)	0.004	97
Height-for-age Z score (SD)	3	1009	776	−0.03 (−0.68, 0.62)	0.94	95
Weight-for-height Z score (SD)	3	1009	776	0.27 (0.09, 0.45)	0.003	66
Weight-for-age Z score (SD)	3	1009	776	0.20 (−0.11, 0.51)	0.21	88
**Pre-Post Studies**	**Number of Studies**	**Total Posttest Participants**	**Total Pretest Participants**	**Mean Difference (95% CI)**	***p*-Value for Summary Effects**	**I^2^ for Heterogeneity (%)**
Hemoglobin concentration (g/L)	7	3370	3817	6.58 (2.71, 10.45)	<0.001	99
Height (cm)	5	2088	2213	2.46 (1.35, 3.58)	<0.001	90
Weight (kg)	5	1800	2212	0.72 (0.33, 1.10)	<0.001	94
Height-for-age Z score (SD)	5	1691	1821	0.24 (0.10, 0.39)	<0.001	70
Weight-for-height Z score (SD)	4	1438	1564	0.28 (−0.05, 0.61)	0.10	95
Weight-for-age Z score (SD)	5	1691	1821	0.25 (−0.05, 0.56)	0.11	95

**Table 4 nutrients-11-02404-t004:** The effect of YYB on prevalence of child anemia and the categorical outcomes of nutritional status in China.

**Post-Only Studies with Concurrent-Control**	**Number of Studies**	**Total Intervention Participants**	**Total Control Participants**	**Risk Ratio (95% CI)**	***p*-Value for Summary Effects**	**I^2^ for Heterogeneity (%)**	**Risk Difference (95% CI)**	***p*-Value for Summary Effects**	**I^2^ for Heterogeneity (%)**
Anemia	11	5869	3518	0.55 (0.45, 0.67)	<0.001	84	−0.13 (−0.20, −0.07)	<0.001	95
Stunting	7	4368	2405	0.60 (0.44, 0.81)	<0.001	71	−0.05 (−0.08, −0.03)	<0.001	52
Underweight	6	2218	1659	0.51 (0.39, 0.65)	<0.001	22	−0.04 (−0.06, −0.03)	<0.001	0
Wasting	5	1831	1419	0.48 (0.32, 0.70)	<0.001	21	−0.02 (−0.04, −0.01)	<0.001	50
**Pre-Post Studies**	**Number of Studies**	**Total Posttest Participants**	**Total Pretest Participants**	**Risk Ratio (95% CI)**	***p*-Value for Summary Effects**	**I^2^ for Heterogeneity (%)**	**Risk Difference (95% CI)**	***p*-Value for Summary Effects**	**I^2^ for Heterogeneity (%)**
Anemia	13	5300	6019	0.58 (0.50, 0.68)	<0.001	89	−0.20 (−0.26, −0.13)	<0.001	93
Stunting	10	3873	4345	0.75 (0.60, 0.95)	0.02	67	−0.03 (−0.05, −0.01)	0.01	68
Underweight	10	3873	4345	0.59 (0.42, 0.83)	0.002	63	−0.03 (−0.05, −0.01)	0.003	72
Wasting	7	2793	3202	0.72 (0.45, 1.15)	0.17	60	−0.01 (−0.03, 0.00)	0.14	64

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
