# Peer review of "The Effect of the Yingyangbao Complementary Food Supplement on the Nutritional Status of Infants and Children: A Systematic Review and Meta-Analysis"

_nutrients, 2019, doi:10.3390/nu11102404_

Round 1

Reviewer 1 Report

The paper is a scientifically sound contribution to the field. Authors followed all the steps for conducting a systematic review and meta-analysis (only a doubt, is it registered in PROSPERO?). English should be carefully revised. 

Author Response

Reviewer 1

The paper is a scientifically sound contribution to the field. Authors followed all the steps for conducting a systematic review and meta-analysis (only a doubt, is it registered in PROSPERO?). English should be carefully revised. 

Response: We did not register the review in PROSPERO. We understand that PROSPERO registration can help preventing duplicate topics. We examined the PROSPERO database and did not identify a similar review of YYB. We have also had a native English speaker edit the manuscript.  

Reviewer 2 Report

This study examines on Yingyangbao, chinese complementary food supplement and its influence on child anthropometry. It is well written and analysis contains large amount of studies. Here are some comments on the manuscript.

General comments:

Is the hematologic status right term to be used about hemoglobin concentration and anemia prevalence? Consider leaving hematologic status out and just writing hemoglobin concentration and anemira prevalence, or replacing it with the some other term, for example: hematological parameters.

Specific comments:

Line 46: ”The base of YYB is full fat powder…” Full fat powder, what, soybean or milk? It should be defined, what the full fat powder is made of, please clarify the sentence. Is the base of YYB always soybean? If so it can be said: the base of YYB is soybean full fat powder…

Lines 49-51: Why to mention multi-micronutrient supplements at all here, as Yingyangbao can be defined s being a complementary food supplement? The sentence could be rephrased as such: Yingyangbao is complementary food supplement providing both micro- and macronutrients and contains calories … and protein…

Table 1:

Yingyangbao is developed to be used by infants and young children aged 6 to 36 months. However in some of the studies the target age has been before 6 months of age. The current age when complementary foods are recommended to be initiated in China should be mentioned somewhere in the manuscript together with the reason, why target age in some of these studies has been before 6 months of age (non-breastfed children etc?).

Why two studies (Yuying  Wang, 2006 and Chunming Chen, 2010) examining YYB association with Developmental quotient are included in the table, as this study examines influence of YYB interventions on growth parameters and hemoglobin?

Check that ”Outcomes measured” are similar in the table. For example at the end of the page 7, Shuai Li, 2017 it says only anemia, should be prevalence of anemia.

Lines 277-302: Why to compare the effect of YYB to multi-micronutrient supplementation, especially when growth parameters are in concern? Concerning weight related indicators, it is not suprising if a supplement containing also macronutrients and energy produces better results than micro-nutrient supplement. Consider discussing more on other similar complementary food supplements than YYB and comparing them with YYB. How do LNS’s differ from YYB, more protein in YYB? Are there similar products than YYB used somewhere? It would be more interesting to discuss more on similar products and what is good (and bad) in YYB compared with these similar products.

Lines 303-309: Irrelevant to this study, as the focus in this manuscript is on anthropometry and hemoglobin. Revise accordingly.

Lines 335-337: Please consider erasing this sentence, is not based on proper reference. As in earlier comment it was said, more discussion on pros and cons of YYB over SIMILAR complementary food supplements are needed for this article.

Author Response

Reviewer 2

This study examines on Yingyangbao, chinese complementary food supplement and its influence on child anthropometry. It is well written and analysis contains large amount of studies. Here are some comments on the manuscript.

General comments:

Is the hematologic status right term to be used about hemoglobin concentration and anemia prevalence? Consider leaving hematologic status out and just writing hemoglobin concentration and anemira prevalence, or replacing it with the some other term, for example: hematological parameters.

Response: We appreciate the reviewer’s comment and have replaced “hematologic status” with “hematological parameters”

Specific comments:

Line 46: ”The base of YYB is full fat powder…” Full fat powder, what, soybean or milk? It should be defined, what the full fat powder is made of, please clarify the sentence. Is the base of YYB always soybean? If so it can be said: the base of YYB is soybean full fat powder…

Response: Yes, it is full fat soybean powder. We have clarified the description of YYB in the manuscript.

Lines 49-51: Why to mention multi-micronutrient supplements at all here, as Yingyangbao can be defined s being a complementary food supplement? The sentence could be rephrased as such: Yingyangbao is complementary food supplement providing both micro- and macronutrients and contains calories … and protein…

Response: We appreciate the reviewer’s comment and have revised the paragraph to be as below (Line 45-51):

“To improve child nutritional status in rural China, Yingyangbao (YYB) was developed as a nutrient-dense complementary food supplement for infants and young children 6-36 months. The base of YYB is full fat soybean powder with additional multiple micronutrients including calcium, iron, zinc, vitamin B1, vitamin B2, vitamin B12, vitamin A, and vitamin D. The composition of YYB varies slightly by the manufacturing company (6–11), and in some formulations folic acid, and omega-3 or omega-6 fatty acids were added. YYB also provides calories (usually around 50 kcal per sachet) and protein (3 g per sachet) (6–8).”

Table 1:

Yingyangbao is developed to be used by infants and young children aged 6 to 36 months. However in some of the studies the target age has been before 6 months of age. The current age when complementary foods are recommended to be initiated in China should be mentioned somewhere in the manuscript together with the reason, why target age in some of these studies has been before 6 months of age (non-breastfed children etc?).

Response: Some earlier studies, particularly before 2007, targeted children from 4 months old due to exclusive breastfeeding recommendations at the time. In 2007, the China Ministry of Health updated the guidelines to recommend exclusive breastfeeding for the first 6 months of life.  We have added this explanation to the manuscript (Line 54-57).

“Early YYB programs provided YYB to children from 4 months old due to national exclusive feeding guidelines that were later changed to 6 months in 2007 (12). Consequently, YYB programs implemented post 2007 were primarily among children 6-36 months old.”

Why two studies (Yuying  Wang, 2006 and Chunming Chen, 2010) examining YYB association with Developmental quotient are included in the table, as this study examines influence of YYB interventions on growth parameters and hemoglobin?

Response: Although this paper mainly focused on growth and hematological parameters, we included reports of YYB on all child health outcomes given most present multiple secondary outcomes. In the “Materials and Method” section, we made it  clear that child developmental outcomes and disease prevalence is also within our consideration, as below (Line 77-86):

 “We sought to examine the effectiveness of providing YYB to infants and children on all health and nutrition outcomes assessed in published studies, including: (1) anthropometric outcomes including height (cm), weight (kg), height-for-age z score (HAZ), weight-for-age Z score (WAZ), weight-for-height Z score (WHZ), stunting prevalence (defined as HAZ≤ -2 standard deviation [SD]), underweight prevalence (defined as WAZ≤ -2 SD), and wasting prevalence (defined as WHZ≤ -2 SD); (2) hematological parameters including hemoglobin (Hb) concentration (g/L) and anemia prevalence (defined as Hb concentration <110 g/L); (3) child developmental outcomes including intelligence quotient and developmental quotient; and (4) disease prevalence including prevalence of diarrhea and prevalence of respiratory infection in past two weeks.”

We also included discussions on the impact of YYB on development quotient and disease prevalence in the Discussion Section (Line 305-311) as below:

“In addition to the effects on hematologic and nutritional outcomes, several papers have examined the association of YYB with additional child health and development outcomes. Two studies that examined developmental outcomes found that YYB program significantly improved intelligence quotient and gross motor development (36,37). YYB contains macronutrients and micronutrients (e.g. protein, iron, zinc, etc.) that may be critical for brain development (38,39). An additional study investigated the effect of YYB on childhood morbidities and found that the prevalence of respiratory infection and diarrhea was reduced after a 12-month YYB intervention (40).  ”

Check that ”Outcomes measured” are similar in the table. For example at the end of the page 7, Shuai Li, 2017 it says only anemia, should be prevalence of anemia.

Response: We have checked “Outcomes measured” in Table 1 thoroughly to make sure they are consistent over studies.

Lines 277-302: Why to compare the effect of YYB to multi-micronutrient supplementation, especially when growth parameters are in concern? Concerning weight related indicators, it is not suprising if a supplement containing also macronutrients and energy produces better results than micro-nutrient supplement. Consider discussing more on other similar complementary food supplements than YYB and comparing them with YYB. How do LNS’s differ from YYB, more protein in YYB? Are there similar products than YYB used somewhere? It would be more interesting to discuss more on similar products and what is good (and bad) in YYB compared with these similar products.

Response: We wanted to present evidence for both multiple micronutrient and LNS since YYB composition is in between these two supplements.  The multi-micronutrient component of YYB is very similar with Sprinkles, Foodlet, and Drops and there is additional energy and protein but it is less than LNS.

 We have revised the Discussion section (Line 280-296) to more clearly present these data. See below:

“Nevertheless, our finding is generally consistent with studies on lipid-based nutrient supplements (LNS), which additionally provide calories, protein, and fatty acids. LNS refers to a range of products where the majority of energy is provided by lipids, including products like Ready-to-Use Therapeutic Foods (RUTF) as well as small-quantity LNS that provide <110 kcal/day.(27). Yet the effect size of YYB on weight-based outcomes in our meta-analysis appear to be much larger than LNS. A recent meta-analysis of RCTs found LNS reduced the risk underweight by 15% as compared to more than a 40% reduction found in our YYB meta-analysis of quasi-experimental studies (27). Similarly, we found YYB had a larger effect on stunting reduction than LNS( 20% as compared to 7%) (27). The difference in the effects of YYB and LNS is notable, particularly considering LNS provides greater calories (usually ~100-1000 kcal per day or 50-100% of daily energy requirement)) as compared to YYB (~50 kcal per day, 5% to 25% of daily energy requirements) (27,31,32).The larger effect of YYB than LNS might be attributable to the differences in the study designs and corresponding risk of bias as well as the study populations. LNS trials were primary for children with moderate acute malnutrition (33), while YYB studies were largely among all children in a community. Therefore, YYB may not only improve growth for undernourished children, but also prevent child malnutrition in the general population, which contributed to a larger effect at population-level.”  

Lines 303-309: Irrelevant to this study, as the focus in this manuscript is on anthropometry and hemoglobin. Revise accordingly.

Response: See the response above.

Lines 335-337: Please consider erasing this sentence, is not based on proper reference. As in earlier comment it was said, more discussion on pros and cons of YYB over SIMILAR complementary food supplements are needed for this article.

Response: We appreciate the reviewer’s comment. We have deleted this sentence from the manuscript. See above for the discussion on YYB with other products.

Reviewer 3 Report

I thoroughly enjoyed reviewing this manuscript titled "Effect of the Yingyangbao Complementary Food Supplement on Nutritional Status of Infants and Children: A Systematic Review and Meta-analysis". For me, it is hard to believe that YYB has such a tremendous impact on hematological and anthropometric parameters but the authors have nicely detailed all the limitations of this systematic review/meta-analysis.

Minor comment:

Page 2, line 84 - Materials and Methods

Can authors explain why "2 weeks" mark was chosen?

Author Response

Reviewer 3

I thoroughly enjoyed reviewing this manuscript titled "Effect of the Yingyangbao Complementary Food Supplement on Nutritional Status of Infants and Children: A Systematic Review and Meta-analysis". For me, it is hard to believe that YYB has such a tremendous impact on hematological and anthropometric parameters but the authors have nicely detailed all the limitations of this systematic review/meta-analysis.

Thank you for the review.

Minor comment:

Page 2, line 84 - Materials and Methods

Can authors explain why "2 weeks" mark was chosen?

Response: The studies only presented 2 week morbidity recalls and therefore we could only present this time period.